# NF-κB inducing kinase is a therapeutic target for systemic lupus erythematosus

Hans D. Brightbill et al.[#]

NF-κB-inducing kinase (NIK) mediates non-canonical NF-κB signaling downstream of multiple TNF family members, including BAFF, TWEAK, CD40, and OX40, which are implicated in the pathogenesis of systemic lupus erythematosus (SLE). Here, we show that experimental lupus in NZB/W F1 mice can be treated with a highly selective and potent NIK small molecule inhibitor. Both in vitro as well as in vivo, NIK inhibition recapitulates the pharmacological effects of BAFF blockade, which is clinically efficacious in SLE. Furthermore, NIK inhibition also affects T cell parameters in the spleen and proinflammatory gene expression in the kidney, which may be attributable to inhibition of OX40 and TWEAK signaling, respectively. As a consequence, NIK inhibition results in improved survival, reduced renal pathology, and lower proteinuria scores. Collectively, our data suggest that NIK inhibition is a potential therapeutic approach for SLE.

. Correspondence and requests for materials should be addressed to N.G. (email: ghilardi.nico@gene.com). [#]A full list of authors and their affiliations appears at the end of the paper.

Systemic lupus erythematosus (SLE) is a chronic auto-immune disease characterized by multi-organ inflammation, resulting from loss of tolerance to self-antigens and production of anti-nuclear antibodies. These antibody–nuclear antigen complexes drive inflammation in multiple organs, including the kidney, resulting in tissue damage[1]. It is thought that nucleic acid–immune complexes activate the innate immune response through Toll-like receptors (TLR) 7 and 9 in plasmacytoid dendritic cells (pDC) and other cell types[2,3], driving production of type I interferon. The resulting interferon signature metric not only correlates with SLE disease severity in human patients[4], but also promotes disease, as blockade of type I interferon signaling through anifrolumab has shown promising efficacy in a phase II clinical trial[5].

In addition to TLR and type I interferon (IFN), several tumor necrosis factor (TNF) receptor superfamily (TNFRSF) members are implicated in SLE pathophysiology[6]. B cell-activating factor (BAFF) and CD40 are required for B cell survival and differentiation to auto-antibody-producing plasma cells[7]. Blockade of CD40 ligand (CD40L) showed promising effects in early lupus clinical trials, even though development was halted due to thrombotic side-effects[8]. On the other hand, BAFF blockade through belimumab is partially efficacious and is the only new therapy for lupus approved in more than 50 years[9]. Furthermore, TNF-related weak inducer of apoptosis (TWEAK) signaling has been implicated in chronic kidney inflammation and disease[10], and a TWEAK blocking antibody is currently in clinical trials for lupus nephritis[11]. Genome-wide association studies have also revealed an expression quantitative trait locus (eQTL) in the gene encoding the T-cell co-stimulatory molecule OX40L, and indeed OX40-L is highly expressed in patients with SLE[12,13]. Furthermore, OX40 agonism accelerates disease, and OX40 blockade delays disease in the NZB/W F1 model of SLE[14]. Combined inhibition of all of these pathways, for example through targeting a common downstream signaling pathway, therefore holds the promise of superior efficacy compared to existing treatments in SLE.

TNFRSF-mediated NF-κB signaling occurs through two distinct pathways. In canonical NF-κB signaling, receptor activation leads to degradation of I-κB downstream of IKK α/β/γ complexes, which results in translocation of canonical NF-κB subunits, such as p65/p50, to the nucleus, where they trigger immune gene expression[15]. Non-canonical signaling is strictly dependent on NIK (MAP3K14)[16]. This pathway is constitutively attenuated by continuous degradation of NIK protein through its association with the TNF receptor-associated factors 2/3-ubiquitin ligase complex[17]. TNFRSF signaling leads to dissociation of NIK from this complex, which allows it to accumulate and phosphorylate IKKα[16]. IKKα in turn phosphorylates NF-κB p100, which then gets cleaved to release the mature transcription factor p52. p52 then dimerizes with RelB, translocates to the nucleus, and triggers transcription of target genes[15,16,18].

NIK deficient mice, or mice that express a variant of NIK protein (aly mutation) unable to associate with IKKα, cannot activate the non-canonical NF-κB pathway. These mutant mice are severely immune-deficient as a result of disrupted splenic and lymphoid organ architecture[19], and reduced or missing lymph nodes[19–21]. These developmental phenotypes are similar to those observed in non-canonical NF-κB (Nfkb2, Relb)-deficient[22], IKKα-AA mutant[16,23], and lymphotoxin or lymphotoxin-β receptor deficient mice[24,25]. NIK deficiency also results in reduced B cell populations similar to BAFF or BAFF receptor 3 (BR3)-deficient mice[22,26,27], reduced germinal center B cell numbers and reduced immunoglobulin (Ig) production[21,26,28]. Similar phenotypes have been detected in patients with mutations in MAP3K14 or NFKB2 that prevent non-canonical signaling[29,30].

Importantly, congenital map3k14 deficiency precludes validation of NIK as a therapeutic target due to the developmental phenotypes it causes. For that reason, we previously generated a conditional allele for map3k14. Deletion in the adult mouse, using a tamoxifen inducible CRE-ERT2 allele, replicated the B cell effects of germline deletion[31], and deletion using CD4-CRE revealed non-redundant functions of NIK in T cells[32]. Furthermore, CD11c-CRE-driven NIK deletion rendered dendritic cells (DC) partially refractory to CD40 stimulation and compromised their cross-priming ability[33]. These results support the idea that pharmacological intervention, using a NIK small molecule inhibitor (SMI), may confer therapeutic benefit with an acceptable safety margin. To test this hypothesis, we generated a potent and highly selective NIK SMI. Using this molecule, here we demonstrate a critical function of NIK kinase activity in non-canonical NF-κB signaling and a disease promoting role in the NZB/W F1 model of SLE[34].

## Results

**NIK SMI1.** NIK SMI1 is a potent SMI of NIK (Fig. 1a, supplementary methods), identified through a program of medicinal chemistry optimization described elsewhere[35]. Docking of the compound to the crystal structure of a previously published analog[36] suggests a very similar binding mode. NIK SMI1 interacts with the hinge region of the ATP-binding site, making hydrogen bonds with Glu472 and Leu474. The alkyne motif traverses a narrow channel, allowing access to a small pocket wherein the ligand makes a number of polar interactions that drive both NIK potency and broad kinome selectivity (Fig. 1b). NIK SMI1 potently inhibits human NIK enzymatic activity (Fig. 1c) and was roughly 1.7-fold less potent on murine NIK (Table 1 and Supplementary Table 2). Inhibition of cellular non-canonical NF-κB signaling was monitored in anti-LTβR-stimulated HeLa cells by high content imaging of nuclear p52 (Fig. 1d) and confirmed by western blot analysis (Supplementary Fig. 1a). NIK SMI1 potently inhibited anti-LTβR-induced nuclear p52 levels in these cells, while it had no effect on canonical signaling as measured by TNF alpha-induced nuclear relA levels (Fig. 1e, Supplementary Fig. 1b). Furthermore, this compound was also highly selective, as it inhibited only 3 out of 222 off-target kinases (KHS1, LRRK2, and PKD1 (PKCμ)) to an extent >75% at a concentration of 1 μM (Fig. 1f, Supplementary Table 1). To better estimate selectivity against these three kinases, we determined the $K_i$ for these enzymes and confirmed NIK SMI1 to be highly selective (Table 1). Taken together, these data establish NIK SMI1 as a unique, highly potent, and highly selective inhibitor of non-canonical NF-κB signaling.

**NIK SMI1 inhibits BAFF and CD40 signaling.** We next investigated the effects of SMI1-mediated NIK inhibition on lupus relevant TNFRSF receptor signaling in primary mouse and human immune cells (Supplementary Table 2). BAFF-induced and CD40-induced p52 processing in human and mouse B cells was significantly inhibited by NIK SMI1 treatment (Fig. 2a, b). We also observed the accumulation of unprocessed p100 protein when CD40 was activated in B cells. This accumulation was most likely a result of p100 induction through the canonical NF-κB pathway downstream of CD40, and thus illustrates selective inhibition of only non-canonical signaling by NIK SMI1. Consistent with this interpretation, BAFF does not induce a strong canonical NF-κB response[16,22,37], and we did not observe p100 accumulation upon BAFF stimulation and NIK inhibition.

NIK SMI1 completely inhibited BAFF-driven survival of mouse and human B cells in vitro with potency on human cells approximately two-fold higher than on mouse cells (Fig. 2c, d).

NIK SMI1 also blocked CD40L-induced ICOSL expression with similar potency as in other assays (Fig. 2e). These data are consistent with observations made in NIK-deficient B cells[38]. In dendritic cells, NIK SMI1 nearly completely inhibited CD40L-induced IL-12p40 production (Fig. 2f), which again is consistent with observations in NIK-deficient DC[33]. NIK SMI1 also reduced anti-CD40-induced IL-12p40 production in WT but not in NIK-deficient mouse splenic DC (Supplementary Fig. 2), illustrating its specific action through inhibition of NIK. NIK-independent readouts, such as anti-IgM or rhCD40L induced B cell proliferation, were not affected (Supplementary Table 2), demonstrating that NIK SMI1 is not generally cytotoxic.

Dose-dependent inhibition of B cell survival was also observed in vivo. C57BL/6 mice were either treated twice daily for 7 days with orally administered NIK SMI1, or with three injections of recombinant BAFF receptor fusion protein (Br3-mIgG2a) over the course of the 7-day experiment as a positive control. NIK SMI1 serum levels at the highest doses exceeded the in vitro determined $IC_{90}$ concentration (Supplementary Fig. 3a), suggesting that complete or near-complete target coverage had been achieved, and correlated well with suppression of marginal zone B cells (Supplementary Fig. 3b). B cell subsets were then monitored by flow cytometry (Supplementary Fig. 3c). NIK SMI1 treatment dose-dependently reduced splenic cellularity (Supplementary Fig. 3d) as well as the frequency and absolute numbers of total B2 B cells, marginal zone B cells, and follicular B cells in the spleen (Fig. 2g–i, Supplementary Fig. 3e-g). Furthermore, NIK inhibition reduced serum IgA levels by approximately 50% after 1 week of treatment (Fig. 2j), as predicted from the phenotype of

NIK deficient mice[31,39–41]. At the highest doses, NIK inhibition quantitatively phenocopied the effects of BAFF blockade, and this is consistent with its non-redundant function in BAFF signaling.

**NIK is required for OX40-induced effects on T cells.** We next investigated whether NIK was critical for OX40 signaling. We first used a conditional allele of NIK crossed to a CD4-driven CRE deleter strain[31,32] and found that anti-CD3/CD28 activated NIK-deficient CD4[+] T cells did not process p52 (Fig. 3a), nor did they hyper-proliferate upon OX40 stimulation (Fig. 3b). Similarly, NIK SMI 1 effectively neutralized anti-OX40 induced hyper-proliferation of splenic CD4[+] memory T cells activated with anti-CD3/CD28 in vitro (Fig. 3c). OX40 stimulation also augmented anti-CD3/CD28 production of IL-2 (Fig. 3d), GM‾CSF, CCL3 and CCL5 (Supplementary Fig. 4) in a NIK dependent manner. Thus, NIK is required for OX40 responsiveness in T-cells, and this result is consistent with T-cell defects observed in *map3k14* [fl/fl] × CD4-cre mice and conventional *map3k14* knockouts[32].

**NIK is required for TWEAK-induced gene expression.** Similar to other TNFRSF members, TWEAK can activate both canonical and non-canonical NF-κB signaling. To address the role of NIK in renal TWEAK signaling, we focused on renal proximal tubulo–interstitial epithelial cells (RPTEC), as this cell type is highly relevant to the well described pro-inflammatory role TWEAK plays in the kidney[42,43]. We first confirmed the requirement for NIK in TWEAK-induced non-canonical NF-κB signaling, and found that p52 processing was significantly

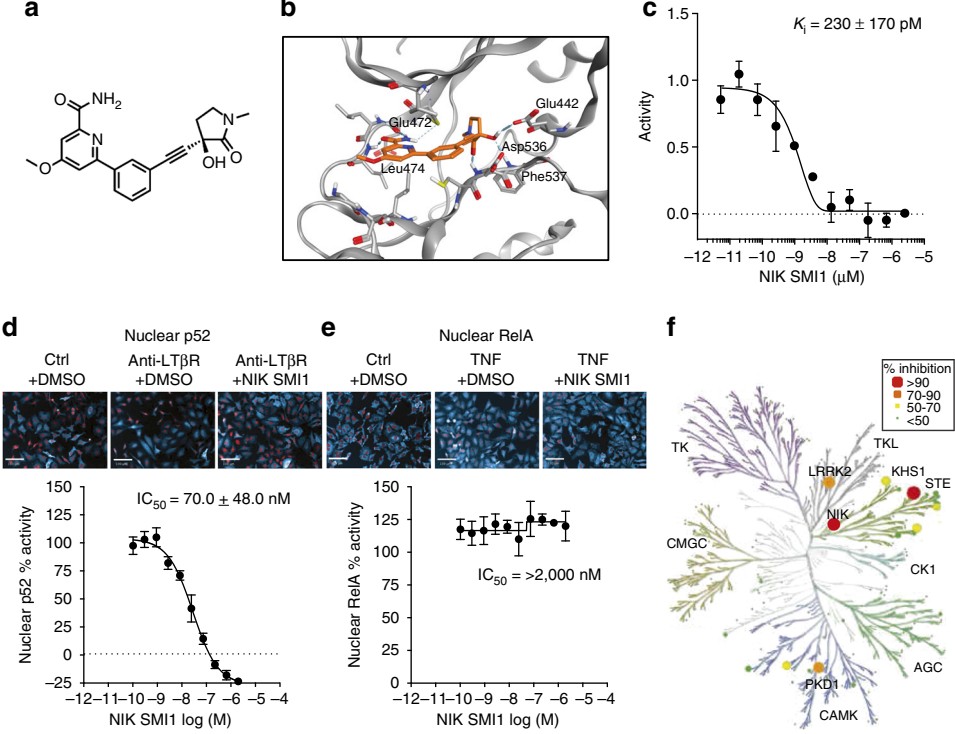

**Fig. 1** NIK SMI1 is a highly selective and potent inhibitor of NIK. **a** Structure of NIK SMI1 **b** Model of NIK SMI1 (orange) docked into the 4G3E crystal structure. The interaction surface around the ligand is depicted in blue and hydrogen bond contacts are in green. **c** Inhibition of human NIK enzymatic activity by SMI1. A representative curve is shown, and the calculated $K_i$ represents the mean ± standard deviation of two independent experiments. **d**, **e** Inhibition of anti-LTβR induced non-canonical (**d**) and TNF induced canonical (**e**) signaling by NIK SMI1. We show representative images of p100/p52 (**d**) or RelA (**e**) in turquoise and nuclear staining for DRAQ5 in red (scale bar = 100 μm), and calculated the titration curves. $IC_{50}$ measurements are listed as mean ± standard deviation from 5–8 independent experiments. **f** Kinase tree illustrating NIK SMI1 kinase selectivity when tested against a panel of 222 kinases. NIK and three potential off-target kinases, KHS1, LRRK2, and PKD1, are marked by colored circles representing percent inhibition observed at 1 μM NIK SMI1

**Table 1 Potency of NIK SMI1 for NIK and most affected off target kinases**

| Target | NIK SMI1 | | | | |
|---|---|---|---|---|---|
| | $K_i$ (nM) | Potency shift vs. hNIK $K_i$ (fold) | Calculated $IC_{50}$ (nM) if ATP = 1 mM | Potency shift vs. hNIK $IC_{50}$ (fold) if ATP = 1 mM | n |
| hNIK | $0.230 \pm 0.170$ | – | $23.2 \pm 17.2$ | – | 2 |
| mNIK | $0.395 \pm 0.226$ | 1.7× | $39.9 \pm 22.8$ | 1.7× | 4 |
| hKHS1 | $49.6 \pm 24.9$ | 216× | $951.4 \pm 477.6$ | 41× | 5 |
| hLRRK2 | $247.8 \pm 175.1$ | 1077× | $3788 \pm 2677$ | 163× | 5 |
| hPKD1(PKCμ) | $75.2 \pm 48.4$ | 326× | $17464 \pm 11204$ | 757× | 3 |

NIK SMI1 $K_i$ were determined in activity assays for the different kinase targets, and predicted $IC_{50}$ at 1 mM ATP for NIK and possible off-target kinases were calculated. Data are represented as mean ± standard deviation of the average for the number (n) of experiments conducted. For details on the calculation, please refer to methods and materials

reduced in murine and human RPTECs stimulated with TWEAK for 4, 8, and 24 h when NIK SMI1 was added (Fig. 4a). Then we stimulated primary murine RPTECs with TWEAK in the presence or absence of NIK SMI1 (3 μM) for 6 and 24 h, and performed RNASeq. Strong induction of gene expression was observed at both 6 and 24 h, and the majority of TWEAK inducible genes were inhibited by NIK SMI1 (Fig. 4b, c). Many of the TWEAK inducible genes fell into three categories of pro-inflammatory genes that are associated with lupus disease, namely cytokines/chemokines, extra-cellular matrix, and signaling genes (Fig. 4d). A subset of these genes was validated by qRT-PCR in mouse and human RPTECs, confirming TWEAK inducibility and NIK dependence (Fig. 4e–g). We also confirmed TWEAK inducibility and NIK dependence for two genes in the human system to ensure that the mouse biology is reflective of the human biology (Supplementary Fig. 5). Of the 179 genes induced by TWEAK and repressed by NIK SMI1 in RPTEC in vitro, 38 genes were found to also be elevated in the kidneys of IFNα-accelerated NZB/W F1 mice with proteinuric disease (Fig. 4h), suggesting the presence of a TWEAK inducible gene signature in diseased kidneys. These genes include pro-inflammatory chemokines and genes involved in pro-inflammatory signaling. Collectively, these data suggest that NIK-mediated TWEAK signaling contributes to kidney pathology in SLE.

**NIK SMI1 suppresses immune responses in vivo.** Based on the broad effects of NIK inhibition in various cell types in vitro, we next tested whether NIK inhibition would be broadly immune suppressive in vivo. First, we tested the effects of a close analog (NIK SMI2, Supplementary Fig. 6, Supplementary Table 2, Supplementary Methods) in an immunization experiment, and found that NIK inhibition reduced germinal center B cell development and antigen-specific IgG1 production, similar to BAFF inhibition (Supplementary Fig. 7). Furthermore, consistent with a previous report[32], NIK SMI1 also affected T-effector/memory cell generation and thus had a partial effect in a delayed type hypersensitivity study (Supplementary Fig. 8). These observations encouraged us to test the effects of NIK inhibition in a disease model.

**NIK SMI1 suppresses disease biomarkers in NZB/W F1 mice.** We next conducted a 4-week biomarker study in NZB/W F1 mice, in which we benchmarked NIK SMI1 to treatment with BR3-mIgG2a (Fig. 5a). At the end of treatment, we evaluated B and T cell subsets by flow cytometry (Supplementary Fig. 9a), splenic and kidney gene expression by Fluidigm RT-PCR, and renal function by proteinuria score. Terminal exposure of NIK SMI1 was found to be at or above the in vitro-determined $IC_{70}$ concentration, suggesting that substantial yet incomplete target coverage over the entire dosing period was achieved

(Supplementary Fig. 9b, c). NIK inhibition did not have a substantial effect on spleen weight or total splenocyte count (Supplementary Fig. 9d, e), but generally replicated the effects of BAFF blockade on B cell-dependent endpoints. Specifically, NIK inhibition decreased the frequency and numbers of splenic B cells, germinal center B cells, and plasma cells, and also led to reduced expression of activation-induced cytidine deaminase (Aid) (Fig. 5b–e, Supplementary Fig. 9 f–h). Interestingly, NIK strongly suppressed expression of Immunoglobulin J (Igj), a plasmablast marker[44] (Fig. 5f), suggesting that it might have a more potent effect on autoantibody production than BAFF inhibition. Indeed, NIK inhibition was superior to BAFF blockade in the suppression of anti-histone (Fig. 5g) and achieved trends in suppression of anti-Sm and anti-ANA (Supplementary Fig. 9i–j).

We next investigated whether the strong effects on plasmablasts and autoantibody production, which were not observed with BAFF blockade, might be due to effects of NIK inhibition on T-cells, perhaps through perturbation of OX40 signaling. In vivo administration of NIK SMI1 to IFNα-accelerated NZB/W F1 mice for 4 weeks did not affect the frequency and numbers of total CD4+ T-cells (Fig. S10a–c), but significantly reduced the frequency and numbers of splenic T effector memory ($T_{EM}$) and T follicular helper ($T_{FH}$) cells (Fig. 6a, b, Supplementary Fig. 10d, e). Consistent with a reduction of $T_{FH}$ cells, NIK inhibition, but not BAFF blockade, reduced expression of ICOS and IL-21 (Fig. 6c, d), which are important T-cell mediators in the cross talk between GC B cell and $T_{FH}$ cells during differentiation[45,46]. We also observed reduced expression of chemokines associated with SLE[47], namely CCL3 (Fig. 6e) and CCL5 (Supplementary Fig. 10f), which we had previously identified as OX40 inducible, NIK dependent genes in T-cells (Supplementary Fig. 4). These data suggest that NIK inhibition potently affects the T cell component of the autoantibody response.

Two potent pro-inflammatory cytokines of myeloid origin, TNF and the p40 subunit of IL-12 and IL-23, were also reduced by NIK inhibition, but not by BAFF blockade (Fig. 6f, g). CXCL16 expression, which has been associated with lupus nephritis and kidney disease severity[48], was significantly reduced with NIK inhibition in the serum (Fig. 6h) of these mice. Several chemokines identified as NIK-dependent and TWEAK-inducible genes in RPTECs (Fig. 4) were reduced by NIK inhibition in the kidney of IFNα-accelerated NZB/W mice, including CCL9, CCL2 (Fig. 6i, j), CXCL13 and CXCL11 (Supplementary Fig. 10g, h). The reduction in chemokines correlated with reduced histopathologic scores of glomerulonephritis (Fig. 6k). We also observed reduced severity of pyelitis, periarteritis and tubulointerstitial nephritis (Supplementary Fig. 10i–k) in mice treated with NIK SMI, but not with BAFF blockade. Proteinuria score was significantly reduced by NIK inhibition after 4 weeks of treatment, whereas BAFF blockade had no effect in this time frame (Supplementary Fig. 10l). Taken

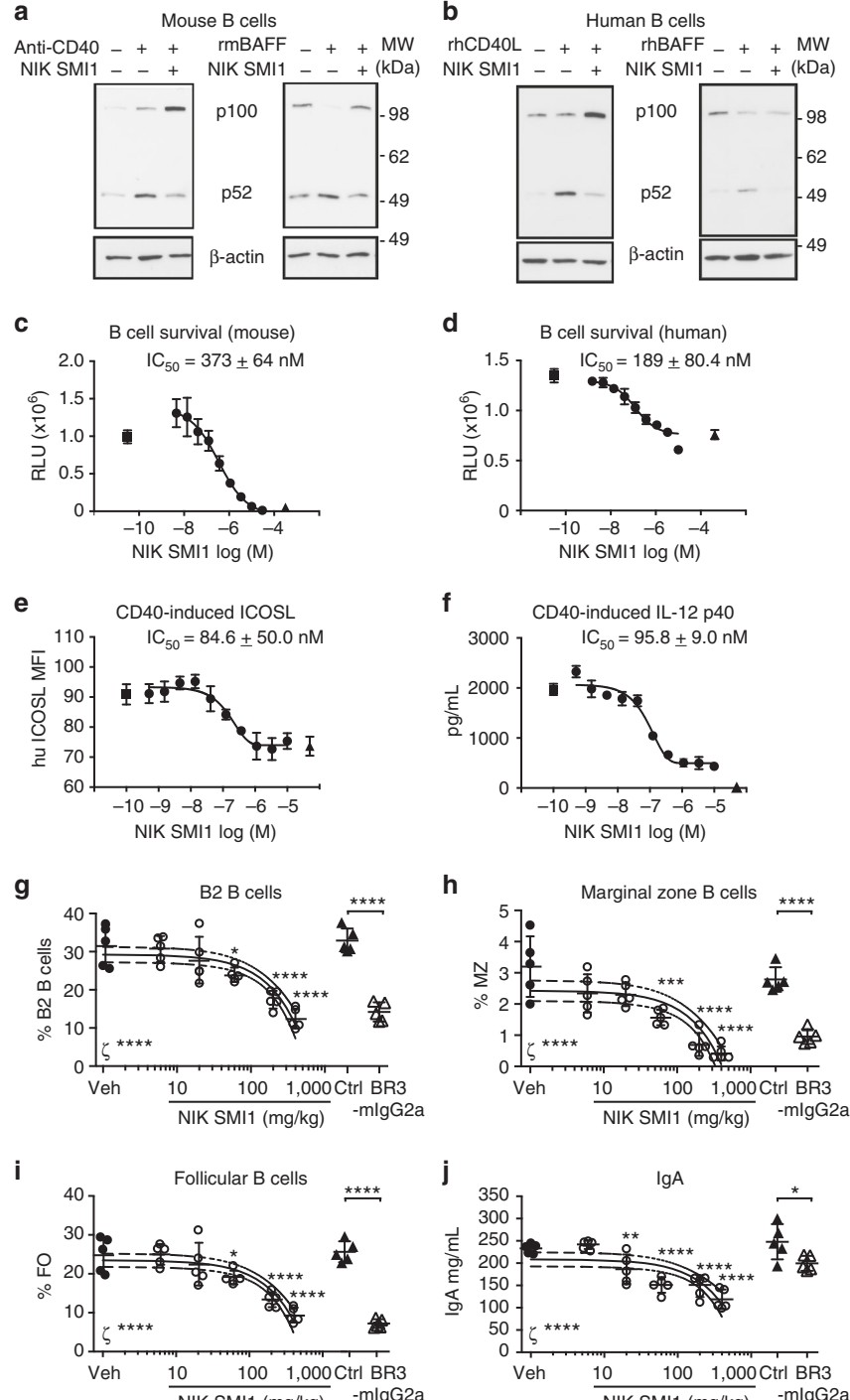

**Fig. 2** NIK SMI1 inhibits BAFF and CD40 induced signaling in B cells and IgA production. **a**, **b** Inhibition of anti-CD40 or BAFF induced p52 processing in murine (**a**) or human (**b**) B cells as analyzed by western blot. **c**, **d** Inhibition of BAFF induced survival of murine (**c**) or human (**d**) B cells in vitro. **e** Inhibition of rhCD40L induced ICOSL expression on human B cells isolated from PBMC. ICOSL MFI was determined by FACS analysis. **f** Inhibition of IL-12p40 production from human monocyte derived DC stimulated with rhCD40L. Triangles, unstimulated + DMSO; squares, BAFF or rhCD40L + DMSO; circles, BAFF or rhCD40L stimulation with NIK SMI1 titration. A representative curve is shown in each case, and IC$_{50}$ values represent mean ± standard deviation of 2 (**c**), 3 (**d**), 4 (**e**), and 3 (**f**) experiments. (**g**–**j**) Dose-dependent in vivo effect of NIK SMI1 on B cell subsets and serum IgA. Frequencies of B2 B cells (CD5$^-$ B220$^+$) (**g**), marginal zone B cells (B220$^+$ CD21$^{hi}$ CD23$^{low}$) (**h**), follicular B cells (B220$^+$ CD21$^{low}$ CD23$^+$) (**i**), and concentration of serum IgA (**j**) are shown. Representative data of three independent experiments are shown. Data represented as mean ± standard deviation of 4–5 mice per group and linear regression curve (solid line) and 95% confidence interval curves (dashed lines) are drawn demonstrating statistically significant (ζ) negative slope and reduction with increasing NIK SMI1 dose. Statistics: Vehicle vs. NIK SMI1 doses, One-way ANOVA with Dunnett's test for multiple comparisons. Isotype control vs. BR3-mIgG2a, unpaired *t*-test with Welch's correction. *$p < 0.05$, **$p < 0.01$, ***$p < 0.001$, ****$p < 0.0001$; ns not significant

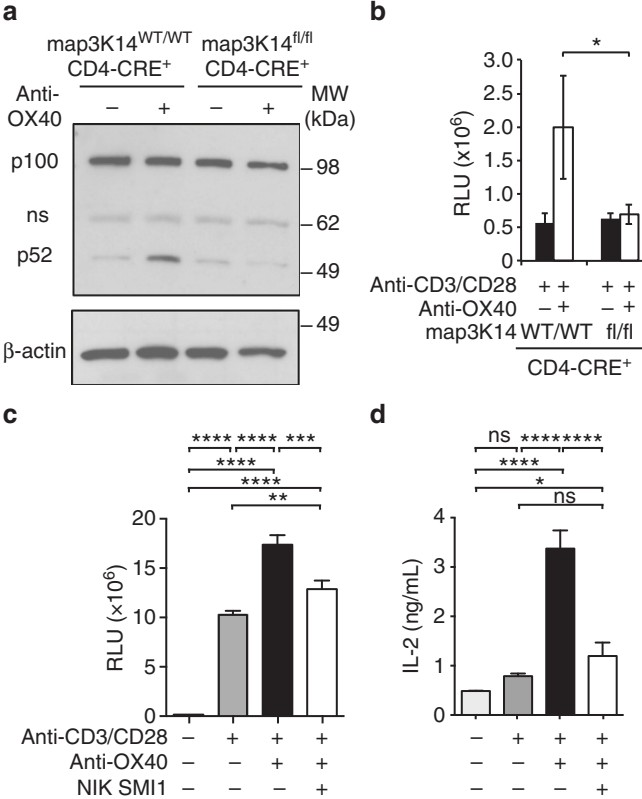

**Fig. 3** NIK is required for OX40-induced T cell co-stimulation. **a** Western blot of anti-OX40 agonist antibody induced p52 processing in anti-CD3/anti-CD28 stimulated CD4[+] T-cells isolated from $Map3k14^{WT/WT}$, CD4-CRE[+] or $Map3k14^{fl/fl}$, CD4-CRE[+] mice. **b** anti-OX40 induced hyper-proliferation of CD4[+] T-cells isolated from $Map3k14^{WT/WT}$, CD4-CRE[+] or $Map3k14^{fl/fl}$, CD4-CRE[+] mice as assessed by Cell Titer Glo ($n = 2$). **c** anti-OX40 induced hyper-proliferation of memory CD4[+] T-cells in the absence or presence of 3 μM NIK SMI1 as assessed by Cell Titer Glo ($n = 2$). **d** Effect of NIK SMI1 on anti-OX40 augmented IL-2 production in **c**. Data are represented as mean ± standard deviation of three biological replicates. Statistics: **b**–**d** one-way ANOVA with Tukey's test for multiple comparisons. *$p < 0.05$, **$p < 0.01$, ***$p < 0.001$; ****$p < 0.0001$; ns not significant

together, our data show that NIK inhibition affects multiple disease relevant pathways in the IFNα-accelerated NZB/W F1 model of SLE.

**NIK inhibition improves survival and renal function.** Given the broad and substantial effects of NIK inhibition on multiple pro-inflammatory endpoints, we next conducted a survival study in IFNα-accelerated NZB/W F1 lupus prone mice. Disease was allowed to progress for 3 weeks after administration of rAdV-IFNα; then mice were treated for 9 weeks with NIK SMI1, BR3-mIgG2a, or appropriate control treatment, and monitored for survival and proteinuria (Fig. 7a). Treatment with NIK SMI1 significantly reduced mortality by about 50% (Fig. 7b). In order to avoid interference with the study to the greatest extent possible, and because we had prior information on PK in the NZB/W model, we conducted only very limited PK sampling in this survival study. Trough concentrations were found to be consistent with the 4-week study, suggesting that substantial yet incomplete target coverage has been achieved (Supplementary Fig. 11). Consistent with previous reports[49], BR3-Fc also improved survival in this model (Fig. 7c). Furthermore, consistent with the reduced renal pathology scores (Fig. 6k, Supplementary Fig. 10i–j) and gene expression patterns (Fig. 4), NIK SMI1-

treated mice exhibited statistically significant reduction in pro-teinuria score, whereas only a trend was observed with BAFF blockade (Fig. 7d, e).

To further validate the claim that NIK SMI1 is efficacious in lupus, we tested the molecule in a second, recently described model, in which lupus is elicited in FVB mice by repeated painting of the ears with a TLR7 agonist[50]. Similar to the data obtained in the IFNα-accelerated NZB/W model, NIK inhibition was efficacious in the survival endpoint (Supplementary Fig. 12). Together, the data from these two efficacy studies demonstrate a clear therapeutic effect of NIK inhibition.

## Discussion

Because genetic ablation of NIK results in developmental defects precluding the establishment of lupus, we chose to validate NIK as an SLE target by generating a highly potent and selective SMI. Using this molecule, we have demonstrated conclusively that disease-relevant, functional effects downstream of the BAFF, OX40, CD40, and TWEAK receptors are critically dependent on NIK kinase activity, even though all of these receptors, with the exception of BAFF-R, also activate NIK-independent, canonical NF-κB signaling. Our findings are based on evidence from in vitro assays specifically tailored to interrogate each individual signaling pathway and their sensitivity to NIK inhibition. To our knowledge, this is the first study defining the functional requirement for NIK kinase activity in the context of OX40, TWEAK, BAFF, and CD40 signaling, and also the first description of NIK as a novel target for lupus.

Like all ATP competitive SMI, NIK SMI1 has limited off-target activities (Table 1, Supplementary Table 1). To minimize interference, we used NIK SMI1 at concentrations corresponding to 90% inhibition of NIK in cellular assays, at which off-target kinases KHS1, LRRK2, and PKD1 (PKCμ) are only minimally affected. Furthermore, we used doses that cover NIK activity to the extent of 70–90% in vivo. Because of the large selectivity window (Table 1), it is highly unlikely that meaningful inhibition of KHS1, LRRK2, or PKD1 (PKCμ) occurred in vivo, and we thus conclude that the observed effects of NIK SMI1 are most likely due to inhibition of NIK kinase activity.

NIK SMI1 has excellent PK properties, which allowed us to examine the role of NIK kinase activity in vivo. Because NIK is downstream of multiple TNFRSF members, it is generally difficult to assign any given in vivo observation to a particular pathway, because all NIK dependent pathways will be inhibited to a similar degree. We initially focused our attention on the BAFF pathway, because it is the only TNFRSF pathway that depends chiefly on non-canonical NF-κB signaling[37] and should therefore be fully inhibited by NIK SMI1, and also because the BAFF blocking antibody Belimumab is an FDA-approved drug for lupus, thus representing an ideal benchmark. Indeed, we found that at high exposure levels, NIK SMI1 phenocopied the effects of BAFF blockade on B cell parameters and serum IgA levels after 1 week of dosing. This experiment not only demonstrated in vivo activity of NIK SMI1, but also suggested a dose level at which complete target coverage was achieved, thus enabling subsequent studies in the lupus disease model.

We used our NIK SMI1 to interrogate the consequences of NIK inhibition in the context of a murine lupus model, IFNα-accelerated NZB/W F1[34]. We chose this model because it is responsive to treatments that are used in human SLE, such as BAFF blockade[49] and cyclophosphamide[51]. We tested NIK SMI1 at a daily dose of 180 mg/kg, which we considered to be an ideal compromise between target inhibition and long-term tolerability. Based on PK sampling, this dose resulted in about 70% inhibition of NIK at the time point of lowest exposure, namely 12 h after the

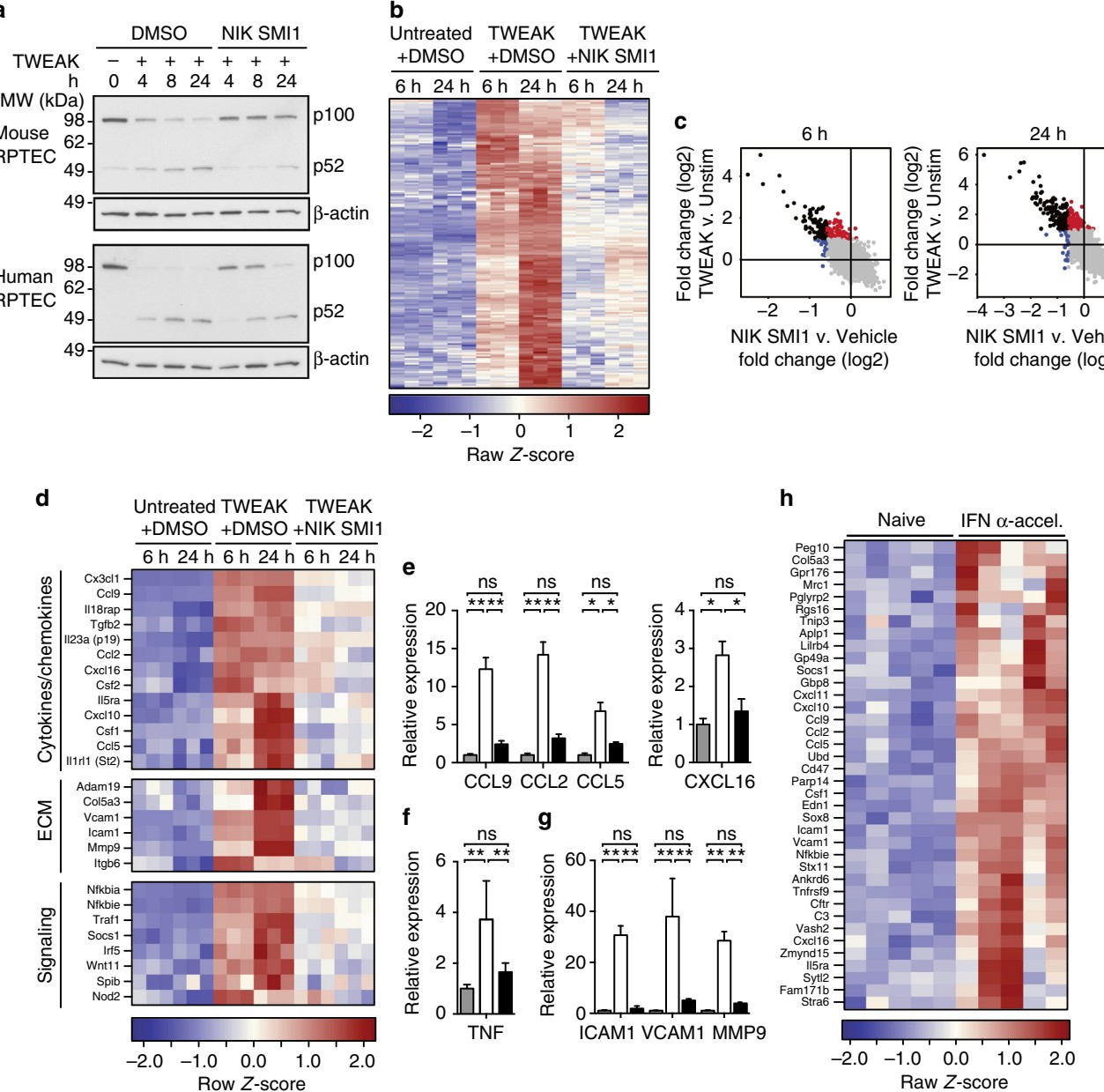

**Fig. 4** NIK is required for TWEAK-induced gene regulation. **a** Western blot analysis of p52 processing in mouse and human renal proximal tubulo-interstitial cells (RPTEC) stimulated with TWEAK in the presence or absence of NIK SMI1 (human 1 μM, mouse 3 μM) at 4, 8, and 24 h. **b–d** Differential gene expression (RNASeq) in murine RPTEC stimulated with TWEAK (100 ng/mL) in the presence or absence of 3 μM NIK SMI1 for 6 or 24 h. **b** Heat map of TWEAK inducible genes (FC > 2, $p < 0.05$) inhibited by NIK SMI1 (FC < −1.5, $p < 0.05$) (**c**) Plot of TWEAK inducible and NIK regulated genes at 6 and 24 h. TWEAK inducible—NIK regulated genes, black; TWEAK inducible—not NIK regulated, red; unaffected by TWEAK, but regulated by NIK, blue. **d** RNASeq heat map of TWEAK inducible—NIK-dependent genes that are specifically relevant to lupus. **e–g** Confirmation by qRT-PCR of select genes identified by RNASeq. qRT-PCR data represented as mean ± standard deviation of three biological replicates run in experimental duplicate, and normalized to expression of HPRT. Unstimulated + DMSO (gray bar), TWEAK (100 ng/mL) + DMSO (white bar), and TWEAK (100 ng/mL) + 3 μM NIK SMI1 (black bar). **h** Heat map, TWEAK inducible—NIK-dependent genes significantly elevated in kidneys of IFNα-accelerated NZB/W F1 mice with proteinuric disease (FC > 2, $p < 0.05$). Statistics: **e–g** one-way ANOVA with Tukey's test for multiple comparisons. *$p < 0.05$, **$p < 0.01$, ***$p < 0.001$; ****$p < 0.0001$; ns not significant

prior dose. Based on more detailed PK profiling (data not shown), this also means that NIK was fully or nearly fully inhibited for the majority of the 12-h dosing interval. While we did not conduct a formal toxicity assessment in mice, we never observed any adverse effect in any of our in vivo studies at that dose that could not be explained by on-target pharmacology. A first analysis after 4 weeks of dosing revealed broad immune suppressive effects of NIK SMI1. Similar to the findings in healthy C57BL/6 mice, NIK

inhibition had profound effects on B cells that were comparable to the effects of the control treatment, BR3-mIgG2a. While we speculate that the effects of NIK SMI1 on B cell subsets are indeed due to inhibition of BAFF signaling, we cannot rule out that NIK also mediates important aspects of CD40 signaling, which is also required for germinal center B cell differentiation and isotype switching[52–55]. Indeed, anti-histone auto-antibodies and expression of the IgJ chain (a plasmablast marker) were more strongly

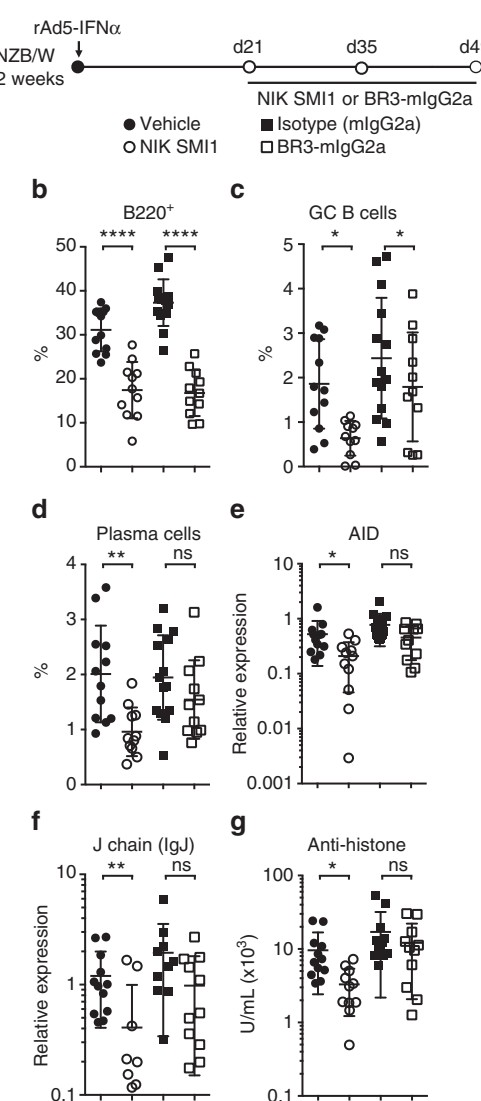

**Fig. 5** NIK inhibition reduces B cell function in NZB/W F1 mice. **a** Experimental design. Lupus prone NZB/W F1 mice were accelerated with adenovirally delivered IFNα and treated with NIK SMI1 or BR3-mIgG2a or appropriate controls for 4 weeks. **b–d** Percentage of B220+ (**b**), germinal center (GC) B cells (B220+ IgM− IgD− GL7hi CD95+) (**c**), and plasma cells (B220− CD138+) (**d**) of live splenocytes. **e, f** Splenic gene expression as determined by Fluidigm technology of AID (**e**) and IgJ (**f**). **g** ELISA measurement of serum autoantibodies against histone. Data represented as mean ± standard deviation from 9–11 mice per group. Statistics: Krushal–Wallis test with Dunn's test for multiple comparisons. *$p < 0.05$, **$p < 0.01$, ***$p < 0.001$; ****$p < 0.0001$; ns not significant

affected by NIK inhibition than by BR3-mIgG2a, and this is likely the consequence of additional pathway inhibition in B cells and beyond.

In addition to BAFF dependent parameters, NIK inhibition also affected T cell parameters associated with SLE, such as expression of ICOS and IL-21. Although our data do not directly establish a causal link, we speculate that the reduction of ICOS and IL-21 expression in vivo is at least partially due to compromised OX40 signaling in T-cells, because both of these genes have previously been described as OX40 regulated genes[56]. Given the prominent function of ICOS and IL-21 in germinal center formation[56–58], and given the eQTL within the OX40L gene linked to SLE[13], we postulate that NIK inhibition in the context of SLE

will lower germinal center formation as a result of inhibition of OX40 signaling in $T_{FH}$ cells. Furthermore, $T_{EM}$ cells are also highly dependent on OX40 signaling and are elevated in human SLE[59,60]. $T_{EM}$ cells were reduced with NIK inhibition in IFNα-NZB/W mice, and a T cell intrinsic requirement for NIK in T cell homeostasis, $T_{EM}$ cell and cytokine responses has recently been demonstrated[32]. Since multiple lines of evidence implicate T-cells, and $T_{FH}$ cells in particular, in human lupus[61], we surmise that the T-cell inhibitory effect of NIK SMI1 will translate to the human system and contribute to disease amelioration in SLE.

Finally, NIK inhibition substantially reduced pro-inflammatory gene expression in IFNα-accelerated NZB/W mouse spleen and kidney. NIK inhibition affected expression of TWEAK responsive genes in the kidney, and mice treated with NIK inhibitors had lower renal pathology scores and less proteinuria compared to control animals. Our in vitro experiments demonstrated that much of TWEAK-responsive gene expression in renal tubulo–interstitial cells is NIK dependent. While NIK SMI1 will likely inhibit more than just TWEAK signaling in vivo, an interpretation in which disease improvement is largely due to inhibition of TWEAK signaling would nevertheless be consistent with a body of literature demonstrating that TWEAK/TWEAKR (Fn14) blockade or deficiency significantly reduce kidney disease in lupus nephritis as well as in other inflammatory preclinical models[62–64].

NIK SMI1 improved survival, not only in IFNα-accelerated NZB/W F1 mice, but also in FVB mice treated with a TLR7 agonist, illustrating that the observed efficacy is not specific to one particular model, but likely broadly applicable to SLE-like disease. However, while the consequences of NIK inhibition were clearly much broader than those of BAFF blockade at the cellular, serological, and histological levels, NIK SMI1 nevertheless fell short of a complete rescue of overall survival. We have several hypotheses that may explain the absence of a more profound therapeutic effect of NIK SMI1: first, as stated above, our PK measurements demonstrated that we did not achieve full target inhibition over the entire dosing period. Although substantial inhibition was achieved, we cannot rule out that a higher dose would have led to more profound therapeutic benefit. Second, the statistical power of a survival study with 15 mice per group is limited, and subtle differences in survival cannot be detected; however, an experiment powered to detect a more limited benefit is not practical. And finally, the IFNα accelerated NZB/W F1 model is well known to be very B cell dependent[65,66]. While our biomarker analysis suggests that pathways like OX40 and TWEAK are active and inhibited by NIK SMI1, it is not known what contribution these pathways make to the mortality end-point. We therefore hesitate to use the survival data as a basis for quantitative predictions of NIK SMI1 efficacy in human SLE.

In summary, we demonstrate that selective inhibition of NIK in vivo leads to inhibition of multiple pathways known to be involved in SLE, resulting in improvement of disease biomarkers, kidney function, and survival. These data position NIK inhibition as potentially superior to BAFF blockade, which has only limited efficacy in human SLE[9]. Given the absence of significant side effects at an efficacious dose, the evaluation of NIK inhibitors in the clinical setting is warranted.

## Methods
**Mice.** Age-matched C57BL/6, NZB/W F1, FVB, Balb/c and CD4-Cre transgenic mice (C57BL/6 background) were purchased from Jackson Labs. Map3k14fl/fl mice and their genotyping have been described previously[31–33]. All animals were bred and housed at Genentech under specific pathogen free conditions. Only female mice were used in these experiments. All animal procedures were conducted under a protocol approved by the Institutional Animal Care and Use Committee at Genentech, and were performed in accordance with the *Guide for the Care and Use of Laboratory Animals*[67].

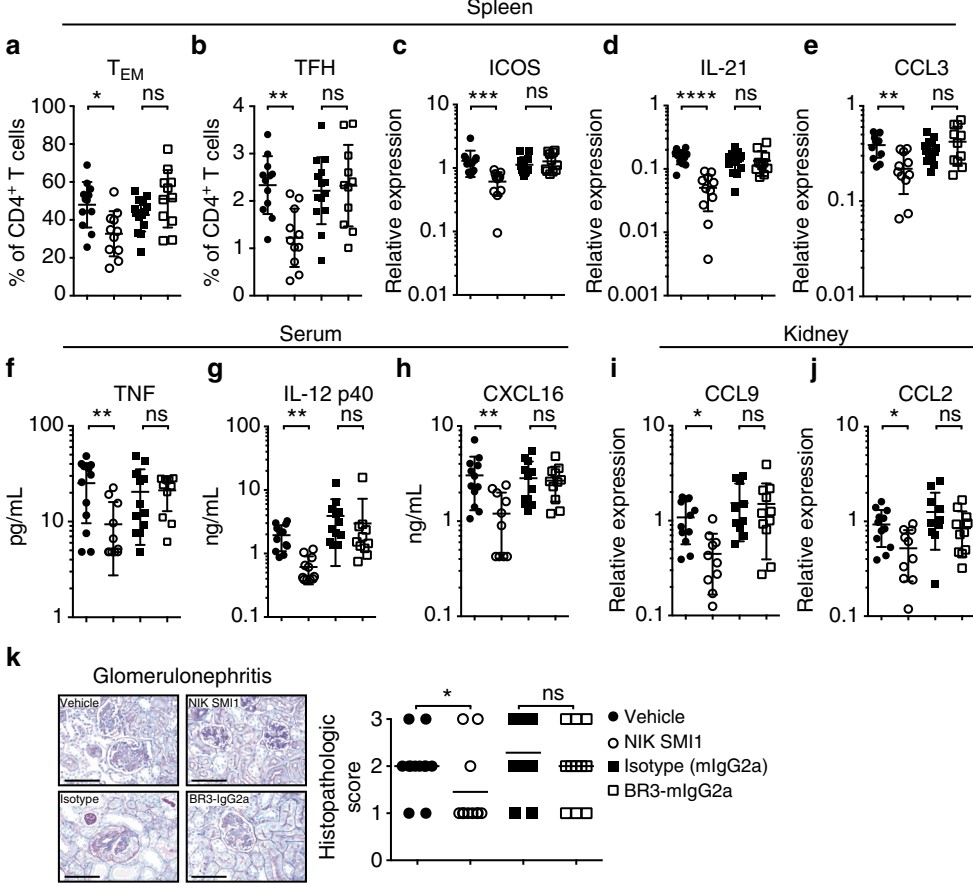

**Fig. 6** NIK inhibition suppresses BAFF-independent biology in NZB/W F1 mice. **a, b** Percentage of T effector memory ($T_{EM}$, $CD4^+$ $CD44^+$ $CD62L^-$) (**a**) and T follicular helper cells ($T_{FH}$, $CD4^+$ $CXCR5^+$ $PD1^+$) (**b**) of $CD4^+$ T cells. **c–e** Splenic gene expression of ICOS (**c**), IL-21 (**d**) and CCL3 (**e**), as determined by Fluidigm. **f–h** Serum levels of TNF (Luminex) (**f**), IL-12p40 (ELISA) (**g**), and CXCL16 (ELISA) (**h**). **i, j** Kidney gene expression, as determined by Fluidigm, of CCL9 (**i**) and CCL2 (**j**). **k** Representative kidney glomeruli (PAS stain) from mice treated with vehicle, NIK SMI1, isotype, or BR3-mIgG2a as annotated (scale bars, 100 μm) and histological scoring. Data in **a–j** are represented as mean ± standard deviation from 9–11 mice per group. Horizontal lines in **k** represent means from 9–11 mice per group. Statistics: **a–j** Krushal–Wallis test with Dunn's test for multiple comparisons. **k** Non-adjusted p-values were generated using Fisher's exact test. *$p < 0.05$, **$p < 0.01$, ***$p < 0.001$; ****$p < 0.0001$; ns, not significant

**Cell culture and human blood cells**. Mouse (C57BL/6) RPTEC (ScienCell) were cultured in EpiCM-a medium with supplements as per vendors protocol. Human renal tubulo–interstitial epithelial cells (Lonza) were cultured in cc-2553 medium with supplements as per vendors protocol. HeLa cells (ATCC) were cultured in Dulbecco's modified Eagle's medium (DMEM) medium (10% fetal bovine serum (FBS), 2 mM Glutamine). All cells were confirmed to be free of mycoplasma. Human immune cells were isolated from healthy human blood donors in accordance with Western Institutional Review Board regulations. Healthy human blood donors were sourced from Genentech employees and gave informed consent for the use of their cells in these studies.

**Reagents**. All antibodies used in this study are listed in supplementary table 3, and all taqman primer/probe sets are listed in supplementary table 4.

**Modeling NIK SMI1 into NIK crystal structure**. The docking of NIK SMI1 to NIK crystal structure was performed using the glide software [https://www.schrodinger.com/Glide]. The extra precision configuration was used with default parameters. The image was created using Molecular Operating Environment, 2014.09[68].

**NIK and potential off-target kinase inhibition assay**. The ability of NIK to catalyze the hydrolysis of adenosine-5′-triphosphate (ATP) was monitored using the Transcreener ADP (adenosine-5′-diphosphate) assay (Bell Brook Labs, Fitchburg, WI). Purified human NIK kinase domain (a.a. 340–694) (0.5 nM) or mouse NIK kinase domain (a.a. 329–675) (1 nM) derived from a baculovirus-infected insect cell expression system was incubated with test compounds for 0.5–2 h in 50 mM HEPES buffer (pH 7.2) containing 10 mM $MgCl_2$, 2 mM dithiothreitol, 10 μM ATP, 0.01% Triton X-100, 0.1% gamma-globulins from bovine blood, 1% dimethylsulfoxide (DMSO), 7 μg/mL ADP antibody and 5 nM ADP-MR121 633 tracer.

Reactions were quenched by the addition of 20 mM EDTA and 0.01% Brij 35. The tracer bound to the antibody was displaced by the ADP generated during the NIK reaction, which caused a decrease in fluorescence polarization that was measured by laser excitation at 633 nm with a Fluorescence Correlation Spectroscopy Plus reader (Evotec AG). Equilibrium dissociation constant ($K_i$) values for NIK inhibitors are calculated from plots of activity vs. inhibitor concentration using Morrison's quadratic equation that accounts for the potential of tight binding, and by also applying the conversion factor that accounted for competitive inhibition and the concentration of substrate used in the assay relative to its Michaelis constant ($K_m$). The $K_i$ values with hPKD1 (PKCμ) were determined using the same assay conditions as for NIK. The $K_i$ measurements for hKHS1 and hLRRK2, as well as broad kinase selectivity profiling for SMI1, were carried out by Invitrogen using their SelectScreen kinase assay panel (ThermoFisher Scientific, Madison, WI). The vendor offered two types of kinase activity assays, a peptide phosphorylation assay (Z′-LYTE) and an ADP quantitation assay (Adapta), as well as one type of kinase ATP site competitive binding assay (LanthaScreen). See vendor web site for technology and protocol details for each specific kinase assay. The ATP concentrations used in the activity assays were typically within 2-fold of the experimentally determined apparent $K_m$ value ($K_m^{app}$) for each kinase, whereas the competitive binding tracer concentrations used in the binding assays were generally within 3-fold of the experimentally determined dissociation constant ($K_d$) values. In Supplementary Table 1, the column (ATP) denotes the concentration of ATP used in each kinase assay ($K_m^{app}$ or 100 μM), while NA (Not Applicable) denotes competitive binding assays that contained no ATP. Finally, theoretical kinase $IC_{50}$ values were calculated assuming the biochemical assays had been run at a more physiological ATP concentration of 1 mM by using the following equation, $IC_{50} = K_i(1 + [ATP]/K_m)$, where $K_i$ is the equilibrium inhibition constant for the inhibitor determined in the biochemical assays, [ATP] is equal to 1000 μM (i.e., 1 mM), and $K_m$ is the Michaelis constant of ATP for the kinase.

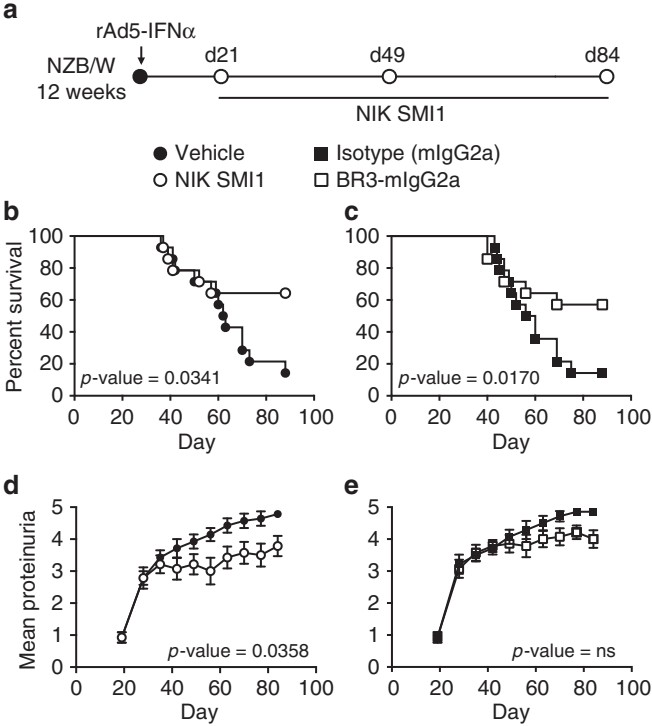

**Fig. 7** NIK inhibition improves survival and proteinuria in NZB/W F1 mice. **a** Experimental design. Lupus prone NZB/W F1 mice were accelerated with adenovirally delivered IFNα and treated with NIK SMI1, BR3-mIgG2a or appropriate controls for 84 days. **b, c** Kaplan–Meier curves of percent survival when treated with NIK SMI1 (**b**) or BR3-mIgG2a (**c**). **d, e** Proteinuria scores of mice treated with NIK SMI1 (**d**) or BR3-mIgG2a (**e**). Proteinuria data are represented as mean ± standard error of the mean of 15 mice for each timepoint. For deceased mice, a proteinuria score of 5 was carried forward. Statistics: survival, Mantel–Cox test; for proteinuria, area under the curve (AUC) from day 19–84 was calculated for each animal, and groups were compared by one-way ANOVA with Sidak's test for multiple comparisons

**High content nuclear p52 and RelA imaging.** The HeLa cell line was used to define the cellular selectivity of NIK inhibitors toward inhibition of classical vs. non-classical NF-κB signaling using high content cellular imaging assays. For the nuclear translocation assay p52 (non-canonical NF-κB signaling), HeLa cells were treated with different concentrations of compounds previously diluted in DMSO (final [DMSO] = 0.2% in all wells) in DMEM medium (10% FBS, 2 mM Glutamine) and then stimulated with 300 ng/mL of an anti-lymphotoxin beta receptor antibody for 4.5 h. In the REL-A nuclear translocation assay, HeLa cells were incubated with compounds for 4 h 50 min before stimulating them with 2 ng/mL TNF (R&D Systems) for 10 min. Cells were fixed with 4% paraformaldehyde, permeabilized by adding 0.1% Triton X-100 in phosphate buffered saline, and then are incubated with either 2 μg/mL anti-p52 antibody or 400 ng/mL anti-REL-A (p65) antibody. Finally, the cells are incubated with an Alexa488-labeled secondary antibody (Invitrogen) and DRAQ5 DNA stain (Biostatus). Imaging was carried out using an Opera Reader (Perkin Elmer) and data were analyzed with Acapella software (Perkin Elmer). The p52 or REL-A translocation into the nucleus is quantified by the ratio of the nuclear to cytoplasmic signal intensity. The concentration of inhibitor required for 50% inhibition (IC50) in these cell assays are derived from the plots of signal vs. inhibitor concentration.

**Kinase selectivity measurements.** The in vitro kinase selectivity of NIK SMI1 was tested at 1 μM in duplicate against 222 recombinant human kinases in the SelectScreen™ kinase panel (ThermoFisher Scientific, Madison, WI). The average % Inhibition of each kinase is reported (Supplementary Table 1).

**Human B cell survival and proliferation assays.** Human B cells were isolated using Miltenyi Human B cell isolation kit following ficoll separation of buffy coats. Cells were re-suspended in RPMI with 10% FBS for the proliferation assays and 2.5% FBS for the survival assays. Mouse B cells were isolated from mouse spleen using Miltenyi mouse B cell isolation kit II. Cells were plated in Co-star 96-well plates at either 50,000 cells/well for the survival assays or at 150,000 cells/well for the proliferation assays. Compounds diluted in DMSO (final DMSO assay

concentration = 0.1%) were added to the cells. The cells were incubated with compound for one hour at 37 °C. Stimulus was then added to the plates and survival or proliferation was measured after four days. For the proliferation assays, cells were treated with either Anti-IgM (20 μg/mL) or rhCD40L (10 μg/mL; R&D) or anti-mouse CD40 (100 ng/mL). Proliferation was measured using thymidine incorporation on day four. For the BAFF survival assay, cells were treated with human or mouse rBAFF (R&D Systems) at 10 ng/ml followed by Cell Titer Glo (Promega) to measure survival on day four.

**ICOSL expression on human B cell assay.** Human B cells were isolated from whole blood. B cells were enriched by RosetteSep protocol (Stem Cell Technologies), and then isolated using Human B cell isolation kit II (Miltenyi). A total of $0.2 \times 10^6$ B cells were cultured and stimulated with rhCD40L (5 μg/mL, 48 h, R&D Systems), or rhBAFF (100 ng/mL, 72 h, R&D Systems) in the presence or absence of a NIK SMI1 or NIK SMI2 dilution series (10 μM − 1.5 nM). Mean fluorescent intensity (MFI) of CD40L and BAFF induced ICOSL expression was measured by flow cytometry on cells stained with anti-CD20 and anti-ICOSL, using a FACS Calibur instrument (BD). Plots and IC50 measurements were generated using GraphPad Prism Software.

**CD40L induced IL-12p40 production from human DCs.** Human monocytes were enriched from whole blood (Monocyte enrichment RosetteSep, Stem Cell) followed by isolation using Monocyte Isolation Kit II (Miltenyi). Isolated monocytes were cultured (RPMI-10% FBS, penicillin/streptomycin, glutamine) with rhGM-CSF (100 ng/mL) and rhIL-4 (6.5 ng/mL) (R&D Systems) for 5 days with GM-CSF replacement on day 3. On Day 5 immature monocyte derived DCs were monitored for differentiation by flow cytometry (CD14+, CD83low CD86low CD209+/hi). Day 5 MDDCs ($0.25 \times 10^6$ per well of a 96-well plate) were then stimulated with rhCD40L (2.5 μg/mL, R&D Systems) with either DMSO or 3-fold serial diluted NIK SMI1 (10 μM – 1.5 nM). After 24 h plates were centrifuged and supernatants were removed for IL-12p40 cytokine measurement by ELISA (BD Biosciences). Plots and IC50 measurements were generated using GraphPad Prism software.

**Agonist anti-OX40 induced T cell costimulation assay.** CD4+ T cells were isolated from CD4-CRE+ *Map3k14*WT/WT and CD4-CRE+ *Map3k14*fl/fl mice using magnetic cell separation technology (Miltenyi). $3 \times 10^6$ (p100/p52 western blot) or 50,000 (T cell co-stimulation assay) CD4+ T cells were incubated with anti-CD3/CD28 beads (1:1 ratio; Invitrogen) for 16 h. Beads were removed and upregulation of OX40 was verified. Cells were then stimulated with agonist anti-OX40 for 8 h (p100/p52 western) or for 3 days (T cell co-stimulation assay as measured by Cell Titer Glo (Promega). Memory CD4+ T cells were isolated from C57BL/6 mouse spleen using an EasySep mouse memory T cell isolation kit (Stem Cell Technologies). 50,000 memory CD4+ T cells were stimulated with anti-CD3/CD28 beads (1:0.5 ratio; Invitrogen) in combination with isotype or agonist anti-OX40 antibody (20 μg/mL; OX-86, eBioscience) and DMSO or NIK SMI1 (3 μM) for 3 days and analyzed by Cell Titer Glo measurement (Promega).

**Antibodies and flow cytometry.** Single cell suspensions of spleen were prepared. Cells numbers were determined by admixture of a known number of fluorescent beads to the cell suspension and read out on a FACS Calibur instrument. FcRs were blocked using mouse FcR Block (Miltenyi). B and T cell subsets were quantified by staining with antibody panels in FACS Buffer (PBS, 2% FBS) for 20 min at 4 °C.

**Western blots.** p52/p100 and β-actin were monitored by western blot as described previously[31]. Briefly, p52 processing was detected after 6 or 18 h of stimulation. Mouse and human RPTEC cells were stimulated with recombinant mouse or human TWEAK (100 ng/mL, R&D Systems). Primary B cells were isolated by negative selection (Miltenyi) from either mouse spleen or human whole blood. Human primary B cells were stimulated with either rhCD40L (10 μg/mL) or rhBAFF (400 ng/mL) (R&D Systems). Mouse primary B cells were stimulated with either anti-CD40 (10 μg/mL, BD Biosciences) or rmBAFF (400 ng/mL). After 6 or 18 h, whole cell extracts were generated using RIPA buffer (with protease and phosphatase inhibition), and approximately 10 μg of total protein was monitored for p52/p100 and β-Actin by western blot. HeLa cells (ATCC) were stimulated for 8 h with anti-LTβR agonist antibody and cytoplasmic and nuclear extracts were generated using kits purchase from Thermo Scientific and approximately 10 mg of each extract was monitored for p52/p100, phospho-RelA, RelA, or β-Actin, Histone H3, or Lamin B1 controls. Uncropped western blots, as required by journal policies, are provided in Supplementary Fig. 13 and 14.

**ELISA and Luminex.** IL-12p40 (BD Biosciences, #555165), CXCL16 (R&D Systems, DY503), IgA (Bethyl laboratories, E90-103), anti-Histone, anti-ANA, and anti-Sm antibodies (Total Ig (A + G + M)) (Alpha Diagnostic International, #5610, #5210, #5405) were measured from mouse serum by ELISA. Serum measurements were performed by 4-point serial dilution (1:2) (as per manufacturer's protocol) starting at 1:50 of serum from IFNα-accelerated NZB/W or C56BL/6 mice treated with NIK SMI, vehicle, BR3-mIgG2a, or isotype control. Luminex measurements of

mouse serum (1:3 dilution) for TNF were done per manufacturer's protocol (Millipore, #MCYTOMAG-70K-PMX-32).

**Seven-day in vivo B cell survival model.** Eight week-old female C57BL/6 mice were dosed orally with vehicle (medium chain triglycerides (MCT)), NIK SMI1 (400, 200, 60, 20, 6 mg/kg, BID for 7 days), or intravenously (i.v.) with isotype control or BR3-mIgG2a (7.5 mg/kg, days 1, 3 and 5). Splenic B cell subsets were analyzed by flow cytometry and serum IgA was monitored by ELISA. Terminal plasma PK measurements of NIK SMI1 were conducted. Animals were randomized into experimental groups upon delivery from vendor. Investigators were not blinded to experimental samples.

**TNP-OVA immunization model.** Four to nine 8-week-old female Balb/c mice were randomized into experimental groups upon delivery from vendor. Mice were dosed with vehicle (MCT, NIK SMI2 (P.O., BID, 300 mg/kg), Isotype-mIgG2a-DANA, anti-CD40L-mIgG2a-DANA (10 mg/kg intraperitoneally (I.P.), days 1, 3, 5 and 8), or BR3-mIgG2a (7.5 mg/kg I.P., days 1, 3, 5 and 8) over the course of 10 days. The naïve group was left unimmunized. Treatment groups were immunized with $TNP_{12}$-OVA (100 µg) (Biosearch Technologies, Inc.) in alum (2 mg, I.P.) (Thermo Scientific) on day 0. Splenic B cell subsets were analyzed by flow cytometry and TNP-OVA specific-serum IgG1 was monitored by ELISA (Genentech, Inc.). Terminal plasma PK measurements of NIK SMI2 were conducted. Investigators were not blinded to experimental samples.

**Delayed-type hypersensitivity model.** Eight female C57BL/6 mice were randomized into experimental groups upon delivery from vendor and treated with NIK SMI1 (BID, P.O., 200 mg/kg) starting on day 1. Mice were immunized with OVA (100 µg, SC) (Sigma) in complete Freund's adjuvant (Spectrum Chemicals and Laboratory Products) on day 0 and then challenged again with OVA (10 µg in PBS, SC right ear) on day 6. Ear thickness was measured by calipers, and unchallenged ear thickness was subtracted from OVA challenged ear thickness after 24 and 44 h. T effector memory cells from the cervical lymph node were monitored by flow cytometry and terminal PK for NIK SMI1 was measured from plasma taken just prior to the end of the study. Investigators were not blinded to experimental samples.

**rAdV-IFNα-accelerated NZB/W F1 lupus model.** Age-matched (NZB x NZW) F1 mice were purchased from Jackson Laboratory. Fourteen to fifteen weeks old female NZB/W F1 mice were injected i.v. with $1.2 \times 10^8$ rAD5-CMV-IFNα (rAdV-IFNα) particles (Genentech) to accelerate disease[34,69]. At 21 days post rAdV-IFNα, animals were randomized prior to experimental treatment with NIK SMI1 (180 mg/kg BID P.O.) or BR3-mIgG2a (7.5 mg/kg, subcutaneously (S.C.), three times per week). In efficacy studies, proteinuria and survival were tracked through day 84. A 4-week PD study was performed to measure cell subsets, serum Ig, chemokines, cytokines, and gene expression.

**Imiquimod-induced FVB mouse lupus model.** Age-matched female FVB mice were purchased from Jackson Laboratory and randomized into experimental groups upon receipt. Ten-week-old female FVB mice were treated with 10 mg of 5% Imiquimod cream (Taro Pharmaceuticals, USA, Hawthorne, N.Y.) applied to the left ear (three doses per week for the duration of the study). Mice were then treated with either NIK SMI1 (200 mg/kg, P.O., BID), Cytoxan (Cyclophosphamide, Baxter Healthcare Corporation) (30 mg/kg, I.P. every 10 days), or vehicle control (MCT). Proteinuria and survival were tracked through day 91. Plasma PK measurement was done at day 85 prior to the end of the study.

**Histology and proteinuria scoring.** Kidneys were taken from IFNα-accelerated NZB/W lupus prone mice after 4 weeks of treatment with NIK SMI1. Kidneys were immersed in 10% neutral buffered formalin at ambient temperature for 24 h, then paraffin-embedded using routine methods. Four micron histologic sections were stained with Periodic acid–Schiff (PAS) for evaluation of glomerulonephritis and hematoxylin and eosin (H&E) for visual evaluation of interstitial inflammation by light microscopy. Scoring by the pathologist was performed in a blinded fashion. Glomerulonephritis severity scoring was on a 4-point semi-quantitative scale: "0" – normal or mild global lesions in <50% of glomeruli, "1"—global lesions in > 50% of glomeruli, <20% of which are severe (defined as >1 segment with <3 patent capillaries), "2"—global lesions in >50% of glomeruli, 20–80% of which are severe, and "3"— >80% of glomeruli with severe global lesions. Chronic pyelitis, cortical periarteritis, and tubulointerstitial nephritis severity was determined by blinded disease scoring. Severity scores for tubulointerstitial nephritis, periarteritis, and pyelitis were all on 4-point semiquantitative scales: "0"—no significant inflammation, "1"—sparse or focal inflammatory infiltrates, "2"—occasional moderately sized inflammatory infiltrates, and "3"—frequent large inflammatory infiltrates. Proteinuria measurements were done by dipstick (Multistix, Bayer Diagnostics). Protein concentration was assigned a proteinuria score with trace = 0.30 mg/dl = 1, 100 mg/dl = 2, >300 mg/dl = 4, and dead mice = 5.

**TWEAK-induced NIK-dependent gene expression in RPTECs.** Murine RPTEC (Sciencell) were cultured as suggested by the vendor. $5 \times 10^6$ cells were stimulated with recombinant mouse TWEAK (100 ng/mL, R&D Systems) for 6 and 24 h in the presence or absence of 3 µM NIK SMI1 (~ mouse $IC_{90}$) or vehicle (DMSO) in 3 replicates per treatment. Equivalent stimulations were also done with human RPTEC cells (LONZA) stimulated with rhTWEAK and 1 µM NIK SMI1 (~human $IC_{90}$) for 6 and 24 h for RT-PCR measurements. Total RNA was extracted using Qiagen RNeasy kit as per manufacturer's protocol including the on-column DNase digestion. Quality control of samples was done to determine RNA quantity and quality prior to their processing by RNA-seq. The concentration of RNA samples was determined using NanoDrop 8000 (Thermo Scientific) and the integrity of RNA was determined by Fragment Analyzer (Advanced Analytical). 0.5 ug of total RNA was used as an input material for library preparation using TruSeq RNA Sample Preparation Kit v2 (Illumina). Size of the libraries was confirmed using 2200 TapeStation and High Sensitivity D1K screen tape (Agilent Technologies) and their concentration was determined by qPCR based method using Library quantification kit (KAPA). The libraries were multiplexed and then sequenced on Illumina HiSeq2500 (Illumina) to generate 30 M of single end 50 base pair reads. Processing and analysis of the RNA sequencing data were performed using the R programming language (http://www.rproject.org) along with packages from the Bioconductor project (http://bioconductor.org). Raw RNA sequencing reads were processed using the HTSeqGenie Bioconductor package. Reads were aligned to the reference mouse genome sequence (National Center for Biotechnology Information build 37) using the Genomic Short-Read Nucleotide Alignment Program algorithm[70]. Uniquely aligned read pairs that fell within exons were counted to estimate individual gene expression levels. For inclusion, genes were required to have >10 reads in at least four samples. To calculate differential expression, we used the DESeq2 Bioconductor package[71] using the default parameters. Data were analyzed and plotted using Expression Plot[72].

**RNA-sequencing from IFNα-accelerated NZB/W.** Previously published RNA-sequencing data from kidneys of IFNα-accelerated NZB/W mice[73] were processed as described above. Kidneys from naïve mice were compared against vehicle-treated mice after onset of disease using DESeq2 with default parameters. Genes were considered differentially expressed if they showed a Benjamini–Hochberg adjusted $p$-value < 0.05, and a fold change >2.

**Fluidigm and qRT-PCR analysis.** cDNA synthesis was performed on 100 ng total-RNA using an iScript cDNA synthesis kit (Biorad). Gene-specific pre-amplification was performed (Applied Biosystems) of target genes and housekeeping genes HPRT1, RPL19 and GAPDH. Fluidigm was performed using the BioMark 96.96 Dynamic Arrays (Fluidigm Corporation) using the manufacturer's protocol. Data were collected using the BioMark Data Collection Software and $C_T$ values were obtained using the BioMark RT-PCR Analysis Software (V.2.1.1, Fluidigm). The relative abundance ($\Delta C_T$) to the geometric mean of HPRT1/GAPDH/RPL19 was calculated: 2^ – (average Ct gene – average $C_T$ geomean [HPRT1/GAPDH/ RPL19]). qRT-PCR analysis was performed using Applied Biosystems 7500 Real time PCR System. The relative abundance ($\Delta C_T$) to HPRT1 was calculated: 2^ – (average Ct gene – average Ct HPRT1). For statistical analyses, values below the lower limit of detection were set to be 10 $C_T$ lower than the lowest recorded value. Genes monitored are listed with primer-probe set numbers in Supplementary Table 4.

**Statistical analysis.** Sample sizes were chosen based on historical experience from other experiments performed in the same models, and maximized to the extent practically possible in order to achieve statistical power. No animals were excluded from analysis in any experiment. Animals were randomized based on weight. Statistical methods were chosen by a professional statistician in consideration of the experimental design, and are described in each figure legend. Generally, comparisons were made between SMI and vehicle treatment, and between isotype and BR3-Fc treatment, but not across all four groups, because the dosing regimens and resulting stress on the animals are sufficiently different to preclude cross treatment comparisons. The data distribution generally met the assumptions of the tests, and variance was often lower in the treatment than in the control groups. In those cases, we used a Krushal–Wallis test with Dunn's test for multiple comparisons. With the exception of histopathology evaluation, experiments were not done in a blinded fashion. Statistical analysis and graphing for each experiment was done using Graphpad Prism 6.0 software.

**Data availability.** Sequence data that support the findings of this study have been deposited in NCBI GEO database with the primary accession code GSE89195. Data referenced in this study are available from NCBI GEO database with the primary accession code GSE72410.

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

## Acknowledgements
We thank Annemarie Lekkerkerker and Leonard Dragone for helpful discussions, and Alberto Gobbi for help modeling NIK SMI1 onto the NIK crystal structure.

## Author contributions
L.C.W. generated the scientific hypothesis and contributed to experimental design. H.D. B. contributed to p52 western blots, BAFF and CD40L-induced ICOSL human B cell cell based assays, CD40-induced p40 in human monocyte-derived DCs assay, OX40-induced T cell co-stimulation assay, performed FACS and ELISAs for all in vivo studies, assisted in mouse experiments, performed data analysis, and wrote the paper, together with N.G. N.R. contributed to fluidigm measurements; E.S. and W.P.L. contributed to IFNα-NZB/ W and TNP-OVA immunization preclinical models. S.S-B. contributed to human and mouse BAFF B cell survival and proliferation cell based assays; Y.C.K. contributed to the Delayed Type Hypersensitivity model. E.B.G. and Z.M. contributed to TWEAK inducible RNAseq and qRT-PCR measurements; B.T.J. and L.C. contributed to CD4-CRE NIK knockout in vitro experiments; S.H. and C.A. contributed to histology; J.D. and J.L. contributed to 7 day B cell survival in vivo PD model; N.B., G.M.C., S.T.S., J.J.C. executed the medicinal chemistry program that led to the identification of NIK SMI1 and NIK SMI2; J.A.H. provided Bioinformatics analysis and guidance for RNAseq experiments; A. R.J., C.E., P.B.K., R.G., S.L. and K.B. contributed to the development and data generation for NIK enzymatic assay and nuclear p52/RelA imaging assays; P.W.F., A.J. and L.M.B. assisted in NIK SMI1 pharmacokinetic (PK) measurements; K.V. provided assistance with mouse breeding and colony maintenance; K.B. provided assistance with NIK compound selectivity measurements and kinase $K_i$ measurements; N.G., C.D.A., M.J.T., M.H.A.I, B.S.M., contributed to experimental design.

## Additional information

**Competing interests:** The authors declare that they are or were paid employees of Genentech, a member of the Roche Group, during the course of these studies, with the exception of S.L., R.G., and K.B. who were contracted through EVOTEC, Inc. All authors with the exception of L.C. and B.T.J. and EVOTEC employees have held equity in the Roche group.

Hans D. Brightbill[1], Eric Suto[2], Nicole Blaquiere[3], Nandhini Ramamoorthi[4], Swathi Sujatha-Bhaskar[1], Emily B. Gogol[1], Georgette M. Castanedo[3], Benjamin T. Jackson[1], Youngsu C. Kwon[2], Susan Haller[5], Justin Lesch[2], Karin Bents[6], Christine Everett[7], Pawan Bir Kohli[7], Sandra Linge[6], Laura Christian[1], Kathy Barrett[7], Allan Jaochico[8], Leonid M. Berezhkovskiy[8], Peter W. Fan[8], Zora Modrusan[9], Kelli Veliz[10], Michael J. Townsend[4], Jason DeVoss[2], Adam R. Johnson[7], Robert Godemann[6], Wyne P. Lee[2], Cary D. Austin[5], Brent S. McKenzie[2], Jason A. Hackney[11], James J. Crawford[3], Steven T. Staben[3], Moulay H. Alaoui Ismaili[7], Lawren C. Wu[1] & Nico Ghilardi[1]

[1]Department of Immunology Discovery, Genentech, 1 DNA Way, South San Francisco, CA-94080, USA. [2]Department of Translational Immunology, Genentech, 1 DNA Way, South San Francisco, CA-94080, USA. [3]Department of Discovery Chemistry, Genentech, 1 DNA Way, South San Francisco, CA-94080, USA. [4]Department of Biomarker Discovery, Genentech, 1 DNA Way, South San Francisco, CA-94080, USA. [5]Department of

Pathology, Genentech, 1 DNA Way, South San Francisco, CA-94080, USA. [6]Evotec, Inc., Essener Bogen 7, Hamburg 22419, Germany. [7]Department of Biochemical and Cellular Pharmacology, Genentech, 1 DNA Way, South San Francisco, CA-94080, USA. [8]Department of Drug Metabolism and Pharmacokinetics, Genentech, 1 DNA Way, South San Francisco, CA-94080, USA. [9]Department of Molecular Biology, Genentech, 1 DNA Way, South San Francisco, CA-94080, USA. [10]Department of Laboratory Animal Resources, Genentech, 1 DNA Way, South San Francisco, CA-94080, USA. [11]Department of Bioinformatics and Computational Biology, Genentech, 1 DNA Way, South San Francisco, CA-94080, USA

