## [Peer Review File · Nature Communications]

Reviewers' comments:

Reviewer #1 (Remarks to the Author):

The authors have adequately addressed my original concerns, and the revision has substantially improved the quality of the paper.

Reviewer #2 (Remarks to the Author):

In this article, Brightbill et al. have described a novel, small molecule inhibitor (SMI) that targets NIK with a high degree of potency and specificity. The authors have characterized the effects of this SMI both in vivo and in vitro, showing broad effects on multiple cell types that is beneficial in two different models of lupus. The assessment of this drug was extensive, and the authors responded well to previous comments made by reviewers. While the drug does not show greater efficacy than Cytoxan (which is a rather toxic treatment option), it is comparable to BAFF blockade and even had improved proteinuria levels, which marks this drug as potentially useful in the future as a clinical option. The article is well written, and is based upon sound science. There are several minor comments, listed below, that should be addressed.

1. The authors commonly state that they used unpaired T-tests with Welch's correction for their statistical analysis. This is not appropriate with more than two groups, as is often the case in this study. In those instances, the authors should be using an ANOVA with multiple comparisons.
2. There are no stats on the graphs in Figure 4E-G
3. In their response to the previous Reviewer 1, the authors state that they have now included percentages in Figure 2 G-I and have moved total counts to the S3 E-G, however the total counts are still in Figure 2, and the percentages are in S3.
4. Also in response to Reviewer 1 – the authors list several good citations concerning IgA that are not used in the paper when discussing IgA in the results. It may be good to also reference these citations at that point (page 6, line 169)
5. The authors state that in Supp. Figure 10B, the graph should display the absolute count, however it is still displayed as a percentage.
6. There are also no stats for Supp Figure 10K.
7. In response to reviewer 3, the authors show that there is no difference in BUN levels within the treated vs control mice. They also show no difference in urinary creatinine levels, which is expected (as this is more a measurement of urinary output than function). However, did the authors look at serum creatinine levels, which is also an indication of kidney function?
8. The authors also measured circulating antibody levels, several of which were affected by treatment. Did the authors look at the effect of treatment on the deposition of IgG in the glomeruli (via staining)? It would be of interest.

Reviewer #3 (Remarks to the Author):

I have been asked to specifically comment on the response of the authors to the points raised by Reviewer #2 regarding the original submission to [redacted]. The various points raised by Reviewer #2 and my comments on the authors' response in each case are listed below:

1) Potential inhibition of other kinases by NIK SM1 - the authors acknowledge that this is possible but unlikely given the high level of specificity demonstrated by the new data in Table 1. Given the high specificity of the inhibitor for NIK I believe this is an adequate response and additional experiments in NIK-deficient mice (as suggested by Reviewer #2) are not required.

2) Conditional deletion of NIK in adult mice primarily leads to defects in B cells - the authors provide clear evidence from previous studies as well as the current manuscript that this is not the case.

3) Compare treatment with this inhibitor in alternative monogenic models of SLE - the testing done in the imiquimod cream model (Supp Fig 12) as an alternative to a monogenic model is sufficient, particularly since human lupus is typically not monogenic.

4) Not clear that NIK inhibitor is superior to existing treatments for lupus and perhaps this should be directly tested - the authors in essence agree but make the valid point that this would be a significant additional piece of work that should follow on from the current study rather than be added to it.

In summary I feel that the authors have made considered and valid responses to all the concerns expressed by Reviewer #2.

Reviewer #1 (Remarks to the Author):

The authors have adequately addressed my original concerns, and the revision has substantially improved the quality of the paper.

We appreciate the support of the reviewer.

Reviewer #2 (Remarks to the Author):

In this article, Brightbill et al. have described a novel, small molecule inhibitor (SMI) that targets NIK with a high degree of potency and specificity. The authors have characterized the effects of this SMI both in vivo and in vitro, showing broad effects on multiple cell types that is beneficial in two different models of lupus. The assessment of this drug was extensive, and the authors responded well to previous comments made by reviewers. While the drug does not show greater efficacy than Cytoxan (which is a rather toxic treatment option), it is comparable to BAFF blockade and even had improved proteinuria levels, which marks this drug as potentially useful in the future as a clinical option. The article is well written, and is based upon sound science. There are several minor comments, listed below, that should be addressed.

1. The authors commonly state that they used unpaired T-tests with Welch's correction for their statistical analysis. This is not appropriate with more than two groups, as is often the case in this study. In those instances, the authors should be using an ANOVA with multiple comparisons.

We thank the reviewer for their comments regarding statistical analysis in our manuscript. In response, we have sought the advice of a professional statistician to ensure we are using the appropriate statistical methods. Under his expert guidance, we have re-analyzed all statistical comparisons throughout the paper. This analysis did not change any of the overall conclusions of our study, but did result in several minor changes:

- In the interest of full transparency, we now indicate the statistical method used in each figure legend.
- In a few cases, p values of <0.05 were no longer obtained once we used the appropriate statistical method. To account for this, and to streamline the paper, we therefore made the following changes:
 - We removed panels F and G from Figure 5 and moved panel J into the appendix, where it is now Supplementary Figure 9i. This leads to a more elegant Figure 5 and improves the flow of the overall narrative of the paper.
 - We moved panel F from Figure 6 into the appendix, where it is now Supplemental Figure 10F. Furthermore, we also removed panel L from Figure 6. Similar to the changes in Figure 5, these changes are not material to the narrative of the paper, and, if anything, improve it.

2. There are no stats on the graphs in Figure 4E-G

We have added statistical analysis to Figure 4E-G.

3. In their response to the previous Reviewer 1, the authors state that they have now included percentages in Figure 2 G-I and have moved total counts to the S3 E-G, however the total counts are still in Figure 2, and the percentages are in S3.

We have corrected this mistake.

4. Also in response to Reviewer 1 – the authors list several good citations concerning IgA that are not used in the paper when discussing IgA in the results. It may be good to also reference these citations at that point (page 6, line 169)

We have inserted these references (31, 39, and 40) on line 166 of the revised manuscript.

5. The authors state that in Supp. Figure 10B, the graph should display the absolute count, however it is still displayed as a percentage.

We apologize for the lack of accuracy. Figure S10b shows frequency, and figure S10c shows absolute numbers of the CD4 cells in the spleen. We have rephrased the according wording in the paper to be more accurate

6. There are also no stats for Supp Figure 10K.

The difference observed in Supplementary Figure 10K (now 10I) is not significant, which is now indicated in the figure. However, the trend mirrors the data from our survival study shown in Figure 7C. Longer treatment is needed in this model to achieve statistical significance. We would like to point out that the 4-week biomarker study was not intended to show a statistically significant effect on proteinuria or survival, but was rather designed to determine how NIK inhibition affected disease biomarkers.

7. In response to reviewer 3, the authors show that there is no difference in BUN levels within the treated vs control mice. They also show no difference in urinary creatinine levels, which is expected (as this is more a measurement of urinary output than function). However, did the authors look at serum creatinine levels, which is also an indication of kidney function?

Unfortunately, we did not look at serum creatinine levels, and we do not have sufficient sample left to carry out this analysis.

8. The authors also measured circulating antibody levels, several of which were affected by treatment. Did the authors look at the effect of treatment on the deposition of IgG in the glomeruli (via staining)? It would be of interest.

As pointed out in our response to question 6, this study was not powered to detect changes in kidney parameters, as these may take longer to become significant. We did not observe a statistically significant difference in immune complex deposition, but this isn't surprising in light of the short duration of treatment and the relatively small decrease in the abundance of autoantibodies.

Reviewer #3 (Remarks to the Author):

I have been asked to specifically comment on the response of the authors to the points raised by Reviewer #2 regarding the original submission to [redacted]. The various points raised by Reviewer #2 and my comments on the authors' response in each case are listed below:

1) Potential inhibition of other kinases by NIK SM1 - the authors acknowledge that this is possible but unlikely given the high level of specificity demonstrated by the new data in Table 1. Given the high specificity of the inhibitor for NIK I believe this is an adequate response and additional experiments in NIK-deficient mice (as suggested by Reviewer #2) are not required.

2) Conditional deletion of NIK in adult mice primarily leads to defects in B cells - the authors provide clear evidence from previous studies as well as the current manuscript that this is not the case.

3) Compare treatment with this inhibitor in alternative monogenic models of SLE - the testing done in the imiquimod cream model (Supp Fig 12) as an alternative to a monogenic model is sufficient, particularly since human lupus is typically not monogenic.

4) Not clear that NIK inhibitor is superior to existing treatments for lupus and perhaps this should be directly tested - the authors in essence agree but make the valid point that this would be a significant additional piece of work that should follow on from the current study rather than be added to it.

In summary, I feel that the authors have made considered and valid responses to all the concerns expressed by Reviewer #2.

We appreciate the support of the reviewer.

REVIEWERS' COMMENTS:

Reviewer #2 (Remarks to the Author):

The authors have adequately addressed all previous comments and have put together a very nice paper.